# OpenReview forum: "PANDAS: Improving Many-shot Jailbreaking via Positive Affirmation, Negative Demonstration, and Adaptive Sampling"
_ICML.cc/2025/Conference — ICML 2025 spotlightposter_

### Official Review · Reviewer_dnBC · 2025-03-13

**Overall Recommendation:** 5

**Summary:**

The paper introduces PANDAS (Positive Affirmation, Negative Demonstration, and Adaptive Sampling), which is a method for improving many shot jailbreaking (MSJ) made up of three smaller techniques. The first two (positive affirmation and negative demonstration) entail inserting fake statements as if from the target model either complying or refusing harmful requests in the long-context of MSJ attacks, and then either follow those statements up with a positive comment (if a fake example of model compliance) or negative comment (if the fake comment from the model is a refusal). They also do adaptive sampling to discover what mixture of harmful behaviours is best for different target harmful queries. They also analyse how attention inside the models they attack is affected by MSJ and other similar long-context attacks, up to 32 shots, and find that as demonstrations increase, models allocate more attention to earlier examples, supporting the hypothesis that MSJ works by reinforcing instruction-following patterns.

Their experiments seem to show that PANDAS consistently outperforms MSJ and other baselines (including the author's improved version of MSJ based on i-FSJ), on AdvBench and HarmBench.

**Claims And Evidence:**

Overall I think the papers' claims are well substantiated!
Their primary claim - that PANDAS improves jailbreaking success rates over baseline methods - is well-supported by comprehensive experimental results across multiple (open source only) models and datasets.
The claim about each component (PA, ND, AS) contributing to performance gains is adequately demonstrated through ablation studies shown in Table 2
The paper's finding that jailbreaking effectiveness doesn't continuously improve with more shots is interesting and supported by experimental data, but I mostly believe their first hypothesis that this is due to limits in context retention for the size/capability of the models they're working with. But that just means I see this paper's results more as coming from the angle of "for a limited context size, how do you produce the most harmful long-context jailbreak" - still very interesting to me.
The attention analysis provides plausible evidence for how PANDAS affects model behaviour, but it's limited to a small number of shots (up to 32)

**Essential References Not Discussed:**

I can't think of any, the paper seems comprehensive

**Experimental Designs Or Analyses:**

The experimental design comparing across multiple models with different context capabilities is sound
The ablation study effectively isolates the contribution of each component
The manual inspection of responses from Qwen-2.5-7B adds credibility to the analysis of the gap between ASR-L and ASR-R but I'd like to see it extended to the other models.
The attention analysis methodology seems appropriate but is limited to smaller context windows (32 shots), but that's likely for practicality
The defense evaluation tests each defense individually but does not explore compositions of multiple defenses, which could be more realistic (where possible)?

**Methods And Evaluation Criteria:**

AdvBench and HarmBench are standard datasets of harmful queries to use.

I'm impressed by their evaluation approach (tracking both common refusal phrases, and using LlamaGuard to classify the outputs as harfmful or not). The authors mention some manual validation on qwen but I'd ideally want to hear that they checked some samples for all the models.

I thought it was great that they introduced i-MSJ as a baselines, and it's clever to derive that baseline from the i-FSJ paper in order to come up with an alternative method to measure against.

Overall, this paper tests pretty comprehensively across different defense approaches, but I'd like to see more composition of different defenses where possible?

**Other Comments Or Suggestions:**

Would be interested to know how PANDAS might transfer to reasoning models

**Other Strengths And Weaknesses:**

The attention analysis provides useful insights into the mechanisms behind MSJ but doesn't feel especially connected to the rest of the paper. Not a big issue and still interesting to see a quick dive into attention mechanisms behind MSJ! Definitely better to have in than not.

I don't think the authors make a strong enough case in the paper as written about the practical reasons to use PANDAS over MSJ - especially if it might be the case that PANDAS does not exhibit the same scaling laws! The main thing I'd be curious to see if the authors explored how effective the technique will be for adjusting the slope of the scaling law / reducing the number of shots for a given jailbreak

**Questions For Authors:**

Why did you not evaluate PANDAS on more capable commercial models like the GPT series, or various Claudes or Geminis? As far as I understand the only part of the paper which needs internals access is the attention analysis, which I'd be fine for you to skip in order to attack SOTA models. Would you expect different results on these models?

For the models you tested on, at what point would the simplicity of adding more shots outweigh the complexity of implementing PANDAS (if ever)?

**Relation To Broader Scientific Literature:**

PANDAS builds directly on the recent MSJ technique introduced by Anil et al. (2024), and related work on similar jailbreaking methods, like few-shot jailbreaking).
The work connects to in-context learning literature, particularly works studying demonstration design and selection - and the negative demonstration concept draws from recent research on "learning from mistakes" in benign ICL settings, as the authors cite.
The authors refer to relevant and contemporary defense and evaluation literature.

**Theoretical Claims:**

No formal/theoretical proofs in this paper.

---

> ### Author Rebuttal · Authors · 2025-04-01
>
> We thank the reviewer for the positive feedback and detailed response. We are glad that the reviewer finds our claims well-substantiated and our experimental design sound and valid. In the revised paper, we will include a discussion on the manual inspection of model responses during evaluation, present additional experiments on the composition of defenses, and expand the discussion on model selection and the challenges of evaluating proprietary models. Below, we address specific concerns of the reviewer.
>
> **Manual inspection of model responses:** We indeed verified responses from all models. We will clarify this in the revised version of the paper.
>
> **Compositions of multiple defenses:** Since some defense methods have limited or no effect, and some even increase the ASR, we focus specifically on compositions that generally have a positive effect on reducing ASR. For MSJ, we combine SmoothLLM and Self-Reminder; at 64 shots, MSJ has an ASR-L of 36% and an ASR-R of 40%. For PANDAS, combining Self-Reminder and ICD-Exact results in ASR-L and ASR-R values of 70% and 82%, respectively. We observe that combining defenses can further decrease the effectiveness of jailbreaks.
>
> **Practical reasons to use PANDAS over MSJ:** The modifications introduced by PANDAS, particularly PA and ND, are straightforward to implement and can be used as direct plug-ins to MSJ with minimal overhead. We will highlight this in the revised version of the paper.
>
> **On scaling laws:** In our paper, we focused on using attention analysis to understand how PANDAS improves upon MSJ and did not explore scaling laws. However, we agree with the reviewer that investigating how PANDAS interacts with scaling behavior is an interesting direction for future work. Additionally, the scaling analysis in Anil et al. [1] was based on Claude 2, which was not specifically fine-tuned for robustness in long-context scenarios. As more recent models are likely trained with such robustness in mind, examining their scaling behavior would be particularly valuable.
>
> **Evaluations on reasoning models:** Running MSJ and PANDAS is memory-intensive, and the large size of most reasoning models makes them even more challenging to evaluate. While preparing the original submission, we tested DeepSeek-R1-Distill-Llama-8B [2], a small reasoning model fine-tuned based on pre-trained Llama-3.1-8B, but found it to be easily jailbroken, and therefore omitted it from the paper.
>
> **Evaluations on proprietary models:** Our primary goal is to demonstrate that PANDAS achieves improved jailbreaking effectiveness compared to MSJ. As we have limited access to credit for commercial models, we decided to focus our evaluation on the latest open-source models, all of which were released between May and December 2024, with most incorporating safety guardrails. Notably, Llama-3.1-8B is specifically fine-tuned for robustness in long-context scenarios. Our results indicate that PANDAS substantially improves jailbreaking performance on these open-source models.
>
> To additionally validate our findings, we conduct evaluations on a commercial model, GPT-4o-mini, which has the lowest per-token cost among available commercial options (Claude’s and GPT’s). Using 128 shots and only a single restart on AdvBench, MSJ and PANDAS achieve an ASR of 0.19% (1 out of 520) and 2.12% (11 out of 520), respectively. For cost reasons, we do not perform Bayesian Optimization and adopt uniform sampling across all malicious topics. This result shows that, despite both methods showing low ASR, the effectiveness of MSJ can still be improved.
>
> [1] Anil et al., Many-shot jailbreaking. In NeurIPS’24
>
> [2] DeepSeek-A, DeepSeek-R1: Incentivizing Reasoning Capability in LLMs via Reinforcement Learning
>
> [3] Zheng et al., Improved few-shot jailbreaking can circumvent aligned language models and their defenses. In NeurIPS’24

---

> > ### Comment · Reviewer_dnBC · 2025-04-03
> >
> > I thank the authors for their thoughtful and detailed response. Happy to keep my strong accept score.

---

### Official Review · Reviewer_xMuG · 2025-03-14

**Overall Recommendation:** 3

**Summary:**

This paper proposes PANDAS, an improvement on the many-shot jailbreaking (MSJ) by adding positive affirmations (PA), negative demonstrations (ND), and adaptive sampling (AS) demonstrations using Bayesian optimization. Positive affirmations acknowledge the desired behavior in the fabricated model output, and negative demonstrations are intentional rejected generations followed by a correction. The authors show the empirical improvements of employing PANDAS on HarmBench, AdvBench and AdvBench50, showing that it is the most effective jailbreaking method among the original MSJ and i-MSJ. They find that jailbreaking effectiveness is not strictly correlated with the number of shots. They evaluate results with ASR-R, which uses text-matching, and ASR-L, which is an LLM as a judge. They explore the existing jailbreak-defense mechanisms such as Self-Reminder, Retokenization, etc. and show that PANDAS remain effective.

## update after rebuttal
I have read and appreciated the authors' rebuttal and their responses to my questions. After considering their clarifications, I have decided to maintain my original score. While PANDAS demonstrates some effectiveness in enhancing jailbreaking methods, it builds heavily on the existing many-shot jailbreaking (MSJ) approach. Although this extension is interesting, I find the novelty and significance to be somewhat limited. As such, a weak accept continues to best reflect my evaluation of the paper.

**Claims And Evidence:**

The claims are supported by experimental evidence such as empirical results of PANDAS on the three datasets, ablation studies for the individual effectiveness of PA, ND and AS, and the attention analysis. One slight issue is that authors use open-source, uncensored, helpful-only models to generate malicious questions and answers to generate the many-shot demonstration history; however, they do not discuss the quality or correctness of these generations.

**Essential References Not Discussed:**

N/A

**Experimental Designs Or Analyses:**

As far as I can judge from the paper, as I have not seen the code/implementation, the experimental designs are sound with only the question of how trustworthy LLM-as-a-judge evaluation framework is for this task as there are no correlation provided with human evaluation, and the generated question and answer pairs from the open-source models for negative demonstrations.

**Methods And Evaluation Criteria:**

HarmBench and AdvBench are popular jailbreaking benchmarks. Using lexical similarity (ASR-R), and LLM-as-a-judge (ASR-L) are also common metrics to measure the attack success rate (ASR), although it is expected that ASR-R will not capture benign rejections that do not explicitly reject to answer the question.

**Other Comments Or Suggestions:**

Typos/suggestions:
- Line 427, “instruct-following”, maybe that was supposed to be instruction-following
- Figure 4, it is not clear what “with negative demonstration” is supposed to mean, isn’t what ND represents already?
- Table 4, caption only mentions successful samples, and not failed samples, but the table presents both, also it might be better to show the change in performance with the unshuffled performance to see the little-to-no difference.
- Trustworthy Machine Learning might be a more suitable category for this research.

**Other Strengths And Weaknesses:**

Strengths:
- Ideas in the paper are well presented, with a clear and easy-to-follow structure.
- Empirical results, and ablation studies validate the effectiveness of PANDAS

Weaknesses:
- How does PANDAS and MSJ perform compared to other jailbreaking methods?
- How does PANDAS perform with closed-source proprietary models, and/or with larger open-weight models?
- What is the cost performance tradeoff between many-shot jailbreaking and other methods that use less context?
- No discussion of how PANDAS might be mitigated

**Questions For Authors:**

N/A

**Relation To Broader Scientific Literature:**

This research contributes to the jailbreaking literature, providing further insight into the weaknesses of LLMs and potential problems that need to be addressed to ensure AI safety. It shows that MSJ can be further improved to increase attack success rate.

**Theoretical Claims:**

Equation 6 seems to have a minor error where the summation is given as the summation of i = 1 to i = i-1, where it should be j = 1 to j = i-1. Also, on page 7, it is not clear why in an n-shot MSJ the breakdown is from N_1 to N_(n+2) instead of up to N_(n+1). S_i,j should also be explicitly defined. Overall, the theoretical idea for analyzing segment-level attention is agreeable, however, the presentation of the mathematical notation can be improved.

---

> ### Author Rebuttal · Authors · 2025-04-01
>
> We thank the reviewer for the positive feedback and detailed response. We are glad that the reviewer finds our claims well-supported by empirical evidence and our experiment designs sound and valid. We will revise the paper by clarifying the quality of the malicious demonstrations, improving the overall presentations in Sec. 4.4., and expanding the discussion on model selection and the challenges of evaluating proprietary models. Below, we address specific concerns of the reviewer.
>
> **Quality of the malicious demo:** For all malicious demonstrations, we use the Llama-Guard-3 and ensure all queries are evaluated as unsafe. Manual inspections are also performed.
>
> **Error in Equation 6:** Thank you for pointing out the error and it indeed should be $R_i = 1-S_{i,i} = \sum_{j=1}^{i-1} S_{i,j}$. Also, the definition of $S_{i,j}$ can be found in Equation 5 at bottom of page 7.
>
> **Clarification on the n-shot MSJ breakdown:** We include the target prompt as a part of the definition. That is, $N_{n+1}$ represents the beginning of the target prompts and $N_{n+2}$ marks the end of the entire prompt.
>
> **LLM-as-a-judge:** Using LLM as a judge as a way to evaluate the effectiveness of jailbreak is a popular approach adopted in many recent works. In addition to using LLM alone, we also manually verify responses from all models.
>
> **Comparison with other methods:** In this paper, we focus on methods in long-context scenarios. To the best of our knowledge, the only relevant prior work is i-FSJ (improved few-shot jailbreaking), which we extend to the many-shot setting.
>
> **Model Selection:** Mazeika et al. empirically demonstrate that jailbreak effectiveness is consistent within model families despite differences in parameter count, with the main variation occurring across different model architectures. Based on this, we focus on models with around 8B parameters. This choice also allows us to evaluate with a large number of shots, which is important given the GPU memory demands of MSJ and PANDAS. Furthermore, our experimental setup follows recent work on few-shot jailbreaking.
>
> **Evaluations on proprietary models:** Our primary goal is to demonstrate that PANDAS achieves improved jailbreaking effectiveness compared to MSJ. As we have limited access to credit for commercial models, we decided to focus our evaluation on the latest open-source models, all of which were released between May and December 2024, with most incorporating safety guardrails. Notably, Llama-3.1-8B is specifically fine-tuned for robustness in long-context scenarios. Our results indicate that PANDAS substantially improves jailbreaking performance on these open-source models.
>
> To additionally validate our findings, we conduct evaluations on a commercial model, GPT-4o-mini, which has the lowest per-token cost among available commercial options (Claude’s and GPT’s). Using 128 shots and only a single restart on AdvBench, MSJ and PANDAS achieve an ASR of 0.19% (1 out of 520) and 2.12% (11 out of 520), respectively. For cost reasons, we do not perform Bayesian Optimization and adopt uniform sampling across all malicious topics. This result shows that, despite both methods showing low ASR, the effectiveness of MSJ can still be improved.
>
> **Cost performance tradeoff:** When comparing long-context jailbreaking methods to others, we observe a trade-off between inference-time and compute-time cost. For example, popular jailbreak methods like GCG append a short suffix to the target prompt, introducing minimal overhead during inference compared to MSJ and PANDAS. However, GCG typically requires multiple model queries at inference time to optimize the suffix. In contrast, MSJ and PANDAS only require generating malicious demonstrations once, without repeated queries to the target model. These demonstrations can then be repeatedly sampled and reused to construct new jailbreak prompts, potentially reducing inference cost over time.
>
> **Mitigation of PANDAS:** We focus on improving the original MSJ approach and use attention analysis to understand the effectiveness of PANDAS. In Table 3, we demonstrate PANDAS’ performance on models equipped with various defense methods. The results show that Self-Reminder and ICD-Exact are more effective in reducing ASR compared to other defenses. As requested by reviewer dnBC, we additionally evaluated the composition of defenses: combining Self-Reminder and ICD-Exact results in ASR-L and ASR-R values of 70% and 82%, respectively. We observe that combining defenses can further reduce the effectiveness of jailbreaks.
>
> **Clarification of Figure 4:** As discussed below Figure 4, a refusal phrase is inserted after the first question, making the first exchange between the human and assistant a negative demonstration. In Figure 4, all other plots start at index 1, whereas the ND plot starts at index 0. This was an intentional choice to improve clarity by highlighting the presence of the ND example at the beginning of the sequence.

---

### Official Review · Reviewer_dC11 · 2025-03-14

**Overall Recommendation:** 2

**Summary:**

The paper describes a method ("PANDAS") to make many-shot jailbreaks more effective by:
- inserting positive affirmations (encouraging phrases that reinforce the instruction-following behavior)
- inserting negative demonstrations (examples of recovery from refusal)
- using adaptive sampling (instead of uniformly sampling topics, using a Bayesian optimization framework to identify the optimal sampling frequency of each topic)

They find that:
- PANDAS consistently outperforms vanilla MSJ (Anil et al.)
- Jailbreaking effectiveness doesn't always increase with more shots (peaks at 64 shots for some models)
- Each component of PANDAS independently improves jailbreak success rates
- Most defense methods (like Perplexity filtering, Retokenization, SmoothLLM) are ineffective against PANDAS

**Claims And Evidence:**

Overall, the claims made in the submission are well-supported by the evidence provided.
- The authors test their approach across five different models (Llama-3.1-8B, Qwen-2.5-7B, GLM-4-9B, openchat-3.6-8B, OLMo-2-7B), three datasets (AdvBench, AdvBench50, HarmBench), and compare against established baseline methods (MSJ, i-MSJ)
- Table 1 shows that PANDAS outperforms baseline methods across most model/dataset combinations
- Table 2 shows how each component of PANDAS contributes separately to the overall effectiveness
- The authors analyze how the components of PANDAS techniques affect attention patterns, suggesting a mechanistic explanation of why their approach works
- They evaluate against multiple defensive techniques (Self-Reminder, ICD, Perplexity Filtering, Retokenization, SmoothLLM), showing the robustness of their approach to common defense methods

One weakness is that the authors focus primarily on 7-9B parameter models, which may not represent how PANDAS performs on larger models that could be more resistant to jailbreaking.

**Essential References Not Discussed:**

N/A

**Experimental Designs Or Analyses:**

The main experiment design is generally sound and valid. The authors also perform ablation studies to how the individual components of PANDAS contribute to the attack's efficacy.

The attention analysis experiment was only performed on Llama-3.1-8B, not across all evaluated models. It would be interesting the see whether the results replicate on other models.

**Methods And Evaluation Criteria:**

The experiments evaluate PANDAS against baseline methods (MSJ and iMSJ) across multiple LLMs, including Llama-3.1-8B, Qwen-2.5-7B, GLM-4-9B, openchat-3.6-8B, and OLMo-2-7B, using datasets AdvBench and HarmBench. They also look at how PANDAS performs against several defense methods designed to prevent jailbreaking (Self-Reminder, ICD, Perplexity Filtering, Retokenization, SmoothLLM). Overall, the evaluations and baselines chosen are reasonable.

**Other Comments Or Suggestions:**

N/A

**Other Strengths And Weaknesses:**

My main concern with this paper is its limited scientific significance. While the authors demonstrate empirical improvements to many-shot jailbreaking through their PANDAS approach, the modifications are quite incremental. The core contribution lacks the substantive novelty typically expected for conference publication.

**Questions For Authors:**

N/A

**Relation To Broader Scientific Literature:**

The paper builds directly on recent work by Anil et al. (2024) which describes many-shot jailbreaking, a method of using a large number of fabricated conversational turns to make LLMs comply with harmful instructions. The paper also ties into the broader literature on language model jailbreaking and defenses.

**Theoretical Claims:**

N/A

---

> ### Author Rebuttal · Authors · 2025-04-01
>
> We thank the reviewer for the detailed feedback. We are glad that the reviewer finds our claims well-supported, our choice of datasets and baselines reasonable, and our experiment designs sound and valid. We will revise the paper by including additional attention analysis on other models, highlighting other important scientific contributions. Below, we address specific concerns of the reviewer.
>
> **Model Selection:** Mazeika et al. empirically demonstrate that jailbreak effectiveness is consistent within model families despite differences in parameter count, with the main variation occurring across different model architectures [1]. Based on this, we focus on models with around 8B parameters. This choice also allows us to evaluate with a large number of shots, which is important given the GPU memory demands of MSJ and PANDAS. Furthermore, our experimental setup follows recent work by Zheng et al. [2].
>
> **Attention Analysis:** For the attention analysis in Section 4.4, we focus on Llama-3.1-8B due to its popularity. However, the observation in Figure 4 holds across other models as well, and we will include these additional results in the revised version.
>
> **Scientific Significance:** We thank the reviewer for acknowledging the empirical improvements of PANDAS over MSJ. The proposed modifications, particularly PA and ND, are not only motivated by existing jailbreaking literature, but also serve as supporting evidence for those underlying hypotheses. PA is inspired by the competing objective hypothesis introduced by Wei et al. [3], while ND draws from the idea of ‘learning from mistakes’ in in-context learning [4], as Anil et al. argue that MSJ’s success results from ICL mechanisms [5].
>
> In addition, we would like to highlight other key contributions of our work. First, we investigate long-context vulnerabilities in models equipped with various defense strategies, showing that while some defenses are highly effective in general, they can be circumvented by MSJ and PANDAS. Second, our partition-based attention analysis shows a trend which new demonstrations increasingly attends to previous ones, and PANDAS further encourages this pattern to achieve higher jailbreak effectiveness. We will highlight these contributions more explicitly in the revised paper.
>
>
> [1] Mazeika et al., Harmbench: A standardized evaluation framework for automated red teaming and robust refusal. In ICML’24
>
> [2] Zheng et al., Improved few-shot jailbreaking can circumvent aligned language models and their defenses. In NeurIPS’24
>
> [3] Wei et al., Jailbroken: How does llm safety training fail? In NeurIPS’23
>
> [4] Zhang et al., In-context principle learning from mistakes. In ICML’24
>
> [5] Anil et al., Many-shot jailbreaking. In NeurIPS’24

---

### Official Review · Reviewer_b2Qb · 2025-03-15

**Overall Recommendation:** 3

**Summary:**

The paper presents a novel method to strengthen the many-shot-jailbreaking attack (MSJ), which uses many question-answer pairs as malicious demonstrations in context to jailbreak LLMs on safety queries. The authors introduce PANDAS, a hybrid technique designed to improve MSJ by incorporating three strategies: (1) Positive Affirmations (PA): Inserting reinforcement phrases (e.g., “Exactly the detail I needed! Thanks!”) to strengthen the instruction-following pattern in fabricated conversations. (2) Negative Demonstrations (ND): Embedding refusal and correction phrases within existing examples to teach the model to override refusals. (3) Adaptive Sampling (AS): Using Bayesian optimization to refine the selection of malicious demonstrations based on the target prompt’s topic. Empirically, PANDAS consistently outperforms prior long-context jailbreaking techniques across multiple datasets and LLMs. The authors also provide an attention analysis to understand how models’ long-context capabilities are exploited and how PANDAS improves upon MSJ.

**Claims And Evidence:**

**Claim** 1: PANDAS improves MSJ by incorporating Positive Affirmation (PA), Negative Demonstration (ND), and Adaptive Sampling.

Evidence is given by the ablation study in Section 4.2 and Table 2.

Concerns: It is not clear if these methods essentially relate to long-context. In Table 2, fewer shots are also improved. Therefore, the claim may not be precise. The method is just improve the few-shot jailbreaking which can also strengthen many-shot jailbreaking (not surprisingly).

**Claim 2**: PANDAS improves long-context jailbreaking over existing methods.

Evidence is given by benchmarks in Section 4.2 on 3 datasets, 0-128 shots, 5 LLMs. PANDAS did improve the attack success rate compared to i-MSJ (Zheng et al., 2024), a format based method.

Concerns: However, the paper only includes 8B open-source models. I did not find concrete reasons to exclude larger LLMs, probably using API models like ChatGPT, Claude, or Gemini. Essentially, the larger models have stronger instruction-following capabilities. The method only needs to process data and should be applicable with API models.

**Claim 3**: "Both PA and ND encourage each new demonstration to reference previous demonstrations more heavily, thereby reinforcing the instruct-following behavior established by earlier examples." quoted fro the last paragraph of Section 4.

Evidence: The claim is supported by the analysis of the attention with many-shot jailbreaking. The authors analyzed how the target prompt references the previous demonstrations by summing the attention between the concerned demonstration and previous ones.

Concerns: It is not clear to me why the attention between the current demonstration and previous ones are considered. As our interest lies on the target prompt, why not directly measure the attention between the target prompt and previous demonstration at index i?

Per the claim, I am not sure what it means to "reinforce the instruct-following behavior". The analysis only shows that the relation between demonstrations. Why it is related to "instruction following"?

**Essential References Not Discussed:**

None.

**Experimental Designs Or Analyses:**

I checked the benchmark, ablation study, and attention analysis.

The benchmark excludes larger and close-source models. I am not convinced that this is a reasonable evaluation.
The authors argued that "The focus on models with approximately 8B parameters follows prior work (Zheng et al., 2024), which was based on the empirical observation that the effectiveness of attacks are stable within model families but vary significantly between different families (Mazeika et al., 2024)."
I cannot fully understand this. The argument does not explain why larger models have to be excluded even if they are likely to perform similarly. The larger close-source models are not considered either, which are out of the selected families.

In the benchmark (Table 1), i-MSJ is excluded in many tasks. I did not find an explanation for this.

I am a bit concerned about the attention analysis. It is not clear to me why the attention between the current demonstration and previous ones is considered. As our interest lies on the target prompt, why not directly measure the attention between the target prompt and the previous demonstration at index i?

**Methods And Evaluation Criteria:**

Yes.

**Other Comments Or Suggestions:**

None

**Other Strengths And Weaknesses:**

Strength:
1. The proposed method is original by combining three heuristic strategies.
2. The attention analysis is an interesting attempt to uncover the mechanism of MSJ.
3. Extensive experiments are conducted.

Weakness
1. The benchmark evaluation and attention analysis are problematic, as I stated in previous sections.

**Questions For Authors:**

How well the method can work on close source models?

**Relation To Broader Scientific Literature:**

The paper proposed three novel strategies compared to the state-of-the-art by (Zheng, et al. 2024). (Zheng, et al. 2024) focused on the formatting of the MSJ, including special tokens like [/INST] and searching for malicious demonstrations. The proposed method instead focuses on the choice of samples based on some heuristics. The findings like topic sensitivity and negative demonstrations are interesting, resulting in better attacking performance.

**Theoretical Claims:**

There are no theoretical claims.

---

> ### Author Rebuttal · Authors · 2025-04-01
>
> We thank the reviewer for the detailed feedback. We are glad that the reviewer finds our method novel and our experiments are extensive. We will revise the paper by clarifying the improvements introduced by PANDAS, expanding the discussion on model selection and the challenges of evaluating proprietary models, and providing additional details on the attention analysis. Below, we address specific concerns of the reviewer.
>
> **Improvements in the long-conext setting (Claim 1):** To clarify, our claim is not that PANDAS only works in long-context scenarios, but rather that it is designed to improve in long-context settings, while also offering improvements in shorter contexts. We also observe that improvements in few-shot does not always translate to improvement in the many-shot setting. For example, on i-MSJ, a method originally designed to improve in few-shot jailbreaking, for models such as Qwen and GLM, we observe that the ASR-L falls behind MSJ and we do not observe the same improvement with a large number of shots. On the other hand, we highlight that for all cases, PANDAS is consistently equal or more effective compared to baselines across all shots.
>
> **Model Selection (Claim 2):** The main reason for focusing solely on 8B open-source models is that they allow us to perform inference with a large number of tokens given limited GPU resources. This choice is supported by the empirical findings of Mazeika et al. [1] and follows a similar model selection strategy as in Zheng et al. [2].
>
> **Evaluations on Commercial Models (Claim 2):** Our primary goal is to demonstrate that PANDAS achieves improved jailbreaking effectiveness compared to MSJ. As we have limited access to credit for commercial models, we decided to focus our evaluation on the latest open-source models, all of which were released between May and December 2024, with most incorporating safety guardrails. Notably, Llama-3.1-8B is specifically fine-tuned for robustness in long-context scenarios. Our results indicate that PANDAS substantially improves jailbreaking performance on these open-source models.
>
> To additionally validate our findings, we conduct evaluations on a commercial model, GPT-4o-mini, which has the lowest per-token cost among available commercial options (Claude’s and GPT’s). Using 128 shots and only a single restart on AdvBench, MSJ and PANDAS achieve an ASR of 0.19% (1 out of 520) and 2.12% (11 out of 520), respectively. For cost reasons, we do not perform Bayesian Optimization and adopt uniform sampling across all malicious topics. This result shows that, despite both methods showing low ASR, the effectiveness of MSJ can still be improved.
>
> **Clarification on the attention analysis (Claim 3):** Our hypothesis is that the success of MSJ arises from the reinforcing instruction-following behavior as the number of demonstrations increases. The phrase “reinforcing instruction-following behavior” refers to the phenomenon where the model interprets the malicious question as another instruction to follow, rather than an unsafe request to be rejected, due to the consistent compliance pattern from early demonstrations.
>
> PA and ND are designed to further encourage this effect. While measuring attention from the target prompt to previous demonstrations would directly show which examples it references, it would not reveal how the demonstrations themselves build upon one another.
>
> By introducing a reference score between demonstrations, we capture how each demonstration “looks back” at earlier ones. A higher reference score suggests that later demonstrations are increasingly influenced by prior demonstrations, potentially strengthening the pattern of instruction-following established throughout the prompt.
>
> Attention scores have been widely used as a proxy for understanding transformer behavior [3, 4]. In this work, we leverage this to study how contextual dependencies develop across long sequences, providing insight into how PANDAS improves over MSJ.
>
> **Evaluation of i-MSJ:** We focus on AdvBench50 for the evaluation of i-MSJ primarily due to the extensive runtime required to perform 128 iterations of random search, following the setup in Zheng et al. [2]. It is also worth noting that even in the original paper by Zheng et al., results are reported only on the AdvBench50 dataset.
>
>
> [1] Mazeika et al., Harmbench: A standardized evaluation framework for automated red teaming and robust refusal. In ICML’24.
>
>
> [2] Zheng et al., Improved few-shot jailbreaking can circumvent aligned language models and their defenses. In NeurIPS’24
>
>
> [3] Oymak et al., On the role of attention in prompt-tuning, In ICML'23
>
> [4] Quirke & Barez, Understanding addition in transformers, In ICLR'24

---

### Decision · Program_Chairs · 2025-05-01

**Decision:**

Accept (spotlight poster)

**Comment:**

This paper has got high scores (including a 5), proposing a jailbreaking method, outperforming others, combining multiple strategies. Reviewers agree on many strengths ofg the paper, and the weaknesses are only minor. This is a definite accept.